# Structural basis for different types of hetero-tetrameric light-harvesting complexes in a diatom PSII-FCPII supercomplex

Ryo Nagao [1,8 ✉], Koji Kato[1,8], Minoru Kumazawa [2], Kentaro Ifuku [3], Makio Yokono [4], Takehiro Suzuki[5], Naoshi Dohmae [5], Fusamichi Akita [1], Seiji Akimoto [6 ✉], Naoyuki Miyazaki [7 ✉] & Jian-Ren Shen [1 ✉]

Fucoxanthin chlorophyll (Chl) *a/c*-binding proteins (FCPs) function as light harvesters in diatoms. The structure of a diatom photosystem II-FCPII (PSII-FCPII) supercomplex have been solved by cryo-electron microscopy (cryo-EM) previously; however, the FCPII subunits that constitute the FCPII tetramers and monomers are not identified individually due to their low resolutions. Here, we report a 2.5 Å resolution structure of the PSII-FCPII supercomplex using cryo-EM. Two types of tetrameric FCPs, S-tetramer, and M-tetramer, are identified as different types of hetero-tetrameric complexes. In addition, three FCP monomers, m1, m2, and m3, are assigned to different gene products of FCP. The present structure also identifies the positions of most Chls *c* and diadinoxanthins, which form a complicated pigment network. Excitation-energy transfer from FCPII to PSII is revealed by time-resolved fluorescence spectroscopy. These structural and spectroscopic findings provide insights into an assembly model of FCPII and its excitation-energy transfer and quenching processes.

[1] Research Institute for Interdisciplinary Science and Graduate School of Natural Science and Technology, Okayama University, Okayama 700-8530, Japan. [2] Graduate School of Biostudies, Kyoto University, Kyoto 606-8502, Japan. [3] Graduate School of Agriculture, Kyoto University, Kyoto 606-8502, Japan. [4] Institute of Low Temperature Science, Hokkaido University, Hokkaido 060-0819, Japan. [5] Biomolecular Characterization Unit, RIKEN Center for Sustainable Resource Science, Saitama 351-0198, Japan. [6] Graduate School of Science, Kobe University, Hyogo 657-8501, Japan. [7] Life Science Center for Survival Dynamics, Tsukuba Advanced Research Alliance (TARA), University of Tsukuba, Ibaraki 305-8577, Japan. [8] These authors contributed equally: Ryo Nagao, Koji Kato. ✉email: nagaoryo@okayama-u.ac.jp; akimoto@hawk.kobe-u.ac.jp; naomiyazaki@gmail.com; shen@cc.okayama-u.ac.jp

Oxygenic photosynthesis is one of the most important biological processes on the earth as it converts solar energy into biologically useful chemical energy and evolves molecular oxygen[1]. To acquire solar energy, photosynthetic organisms have developed various types of light-harvesting complexes (LHCs) to absorb light energy under different living environments and to transfer them to two multi-subunit membrane protein complexes, photosystem I (PSI) and II (PSII), to drive a series of charge-separation and electron transfer reactions. The protein subunits and cofactors in the core of PSII and PSI are largely conserved among oxyphototrophs; however, LHCs differ remarkably in different organisms. This is due to varying compositions of chlorophylls (Chls) and carotenoids (Cars) that are bound to LHCs, as well as protein sequences, leading to the color variations of the organisms[2].

Based on the color variations, oxyphototrophs can be divided into green and red lineages[2]. The structures of supercomplexes of PSII in complex with PSII-specific LHCs (PSII-LHCII) from the green-lineage organisms have been determined by cryo-electron microscopy (cryo-EM) single-particle analysis from land plants[3,4] and green algae[5,6]. These structures showed that LHCII is mainly organized into trimers, and two or three LHCII homo- and/or hetero-trimers are associated with each PSII core, in addition to several LHC monomers. In the red-lineage organisms, cryo-EM structures of unique supercomplexes of PSII with fucoxanthin (Fx) Chl *a*/*c*-binding proteins (FCPs), namely PSII-FCPII, have been solved from the diatom *Chaetoceros gracilis*[7,8]. It was found that FCPII is organized into tetramers, and two FCP homo-tetramers, one is strongly-associated (S-tetramer) and the other one is moderately-associated (M-tetramer), are bound to each PSII core together with three FCP monomers. These distinctive differences of LHC organizations with different pigment compositions between the green- and red-lineage organisms appear to result from adaption to the different light conditions to which each lineage organism is exposed[9,10].

Diatoms contain a large number of FCP genes and subunits; for example, the genome of *C. gracilis* (ChaetoBase; https://chaetoceros.nibb.ac.jp/) shows 46 FCP genes[11]. Unlike the green-lineage PSII-LHCII supercomplexes, some of the FCPII subunits in the structures of diatom PSII-FCPII supercomplexes reported previously have not been identified yet[7,8]. One of the previous PSII-FCPII structures was reported at a 3.8 Å resolution[7], which could not assign the three types of monomeric FCP subunits or the characteristic pigment molecules of Chl *c* and diadinoxanthin (Ddx).

On the other hand, the other report showed the PSII-FCPII structure at a resolution of 3.0 Å[8]; however, not all of the FCPII subunits were assigned because of the significantly lower resolution in the peripheral regions where some FCPII subunits are located. In addition, the sequences of the S-tetramer and M-tetramer differ considerably between the two studies[7,8]. Another problem is that transcriptome data were used in the latter study to obtain the FCP genes[8]. Because of the high similarity among FCP genes, it is difficult to reconstruct the whole set of FCP genes from short-read RNA sequences. Thus, the identity of each FCPII subunit in the PSII-FCPII supercomplex remains to be elucidated.

In this study, we improved the resolution of the PSII-FCPII supercomplex structure remarkably to 2.5 Å by cryo-EM single-particle analysis. As a result, all FCPII subunits, together with most Chls *c* and Ddxs, were assigned based on the high-resolution maps obtained. The structure revealed two types of FCP hetero-tetramers and identified a highly complicated pigment network involving Chl *a*, Chl *c*, Fx, and Ddx, which is important for light energy-harvesting, transfer, and quenching.

## Results

**Overall structure of the PSII-FCPII supercomplex.** Cryo-EM images of the PSII-FCPII were obtained by a Titan Krios electron microscope. The final density map of the PSII-FCPII (named $C_2S_2M_2$ here, where $C_2$ indicates a PSII core dimer, and $S_2$ and $M_2$ indicate two 'strongly' and two 'moderately' associated FCP tetramers, respectively) was obtained at a resolution of 2.5 Å with a C2 symmetry upon data processing of the resultant images by RELION (Supplementary Fig. 1 and Supplementary Table 1), based on the "gold standard" Fourier shell correlation (FSC) = 0.143 criterion (Supplementary Fig. 2). This PSII-FCPII also contains six FCP monomers, namely two sets of monomer1 (m1), monomer2 (m2), and monomer3 (m3), as described previously[7]. The resultant map of PSII-FCPII has weak densities in the peripheral region, especially in m3; therefore, we performed focused-3D refinement of the peripheral FCPII ($S_1M_1$; Supplementary Fig. 1). The final density map of the $S_1M_1$FCPII was obtained at a resolution of 2.8 Å with a C1 symmetry (Supplementary Figs. 1, 2). The atomic model of PSII-FCPII was built based on these two types of density maps (Fig. 1).

The structure of the PSII core is basically similar to the previous structures of PSII-FCPII[7,8], which contains 19 membrane-spanning subunits (D1, CP47, CP43, D2, PsbE, PsbF, PsbH, PsbI, PsbJ, PsbK, PsbL, PsbM, PsbT, PsbW, PsbX, ycf12, PsbZ, Unknown0, and

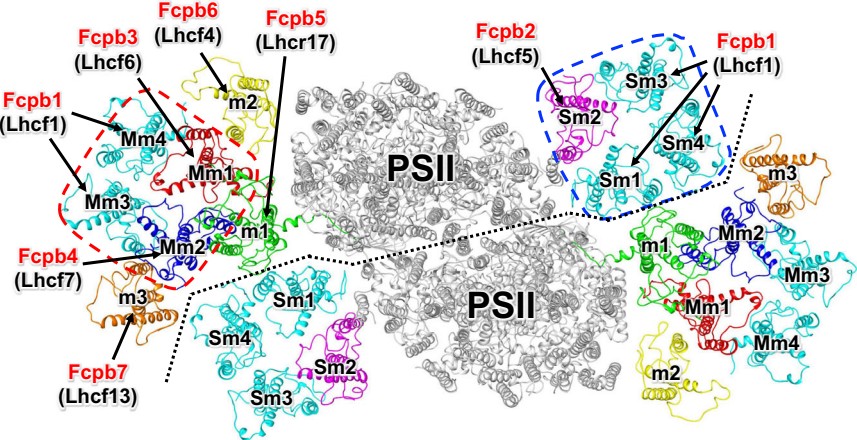

**Fig. 1 Overall structure of the PSII-FCPII supercomplex.** Overall structure of the PSII-FCPII supercomplex viewed from the stromal side. PSII cores are colored gray, whereas FCPII is labeled and colored differently. Each FCP subunit is annotated as Fcpb1–7 (red) with their gene products indicated in parentheses (black). Blue and red dashed boxes indicate S and M-tetramers, respectively. Black dotted line stands for the interface between the two PSII-FCPII monomer units.

Unknown1) and 4 extrinsic subunits (PsbO, PsbQ′, PsbV, and PsbU) (Supplementary Fig. 3). Both Unknown0 and Unknown1 could not be identified yet, even in the 2.5 Å resolution structure. On the other hand, Unknown2 observed previously[7] was assigned to an N-terminal motif of PsbQ′. The extensive N-terminal region of PsbQ′ was already identified in the structure reported previously[8]. In addition, the fifth extrinsic subunit, Psb31[12–15], is not found in the present structure, although it is present in one of the PSII-FCPII structures reported previously[8]. The binding site of Psb31 in the previously reported structure[8] is consistent with suggestions from our biochemical studies, including cross-linking, release-reconstitution, chemical modifications, and site-directed mutagenesis[13,16–18].

All FCPII subunits were assigned in this study, and their structures are well fitted to the densities of each FCPII subunit (Supplementary Fig. 4 and Supplementary Table 2). In the previous studies, the S-tetramer and M-tetramer were tentatively assigned to homo-tetramers composed of the gene product of Lhcf1[7] or FCP-A, which is equivalent to the gene *lhcf8* from the diatom *Thalassiosira pseudonana* with some ambiguities[8]. Each monomer in the S-tetramer is named Sm1, Sm2, Sm3, and Sm4, and each monomer in the M-tetramer is named Mm1, Mm2, Mm3, and Mm4 (Fig. 1). In this study, improvement in the resolution of the cryo-EM map enables us to determine that Sm2, Mm1, and Mm2 are the gene products of Lhcf5, Lhcf6, and Lhcf7, respectively, whereas the remaining ones are Lhcf1 (Fig. 1). This indicates that the S-tetramer and M-tetramer are different types of hetero-tetramers. In addition, three monomers, m1, m2, and m3, are identified as the gene products of Lhcr17, Lhcf4, and Lhcf13, respectively (Fig. 1). Here, we name the FCPII subunits as Fcpb1–7 (Fig. 1), and their assignment was summarized in Fig. 1 and Supplementary Table 4.

The root mean square deviations of Sm1 with each of the 10 FCPII subunits are ranged in 0.42–2.06 Å for the Cα atoms (Supplementary Table 4). The present resolution of the cryo-EM map allows us to identify most of the Chl *c* and Ddx in FCPII. As a result, the pigment ratios of Chl *a*:Chl *c*:Fx:Ddx are 26:14:24:0 in the S-tetramer; 23:15:22:1 in the M-tetramer; 9:1:2:1 in m1; 6:3:2:2 in m2; and 7:2:4:2 in m3. This indicates that in comparison with the amount of Chl *a*, the amount of Fx is much higher, and that of Ddx is much lower in the two tetramers than the three monomers. The significantly higher amount of Fx relative to Chl *a* found in the two tetramers is similar to those found in the structure of an isolated FCP dimer[19]. On the other hand, the amount of Chl *c* relative to Chl *a* is also much higher in the two tetramers than the two monomers except m2. These differences in pigment contents appear to be related to the unique light-energy harvesting, transfer, and quenching properties of each FCPII subunit. The whole FCPII associated with each PSII core contains 71 Chls *a*, 35 Chls *c*, 54 Fxs, and 6 Ddxs (Supplementary Table 3), in comparison to 106 Chls *a* and 59 Fxs[7] or 77 Chls *a*, 29 Chls *c*, 62 Fxs, and 1 Ddx[8] reported previously.

**Structure of the S-tetramer**. The S-tetramer is composed of three Fcpb1s (Sm1, Sm3, and Sm4) and one Fcpb2 (Sm2) (Fig. 1). Sm1-Fcpb1 has 4 Chls *c* and 6 Chls *a* in addition to 6 Fxs (Fig. 2a and Supplementary Table 3). There are two dimeric Chls in Sm1-Fcpb1: one is a homo-dimeric c302/c306 with an edge-to-edge distance of 3.7 Å, and the other is a hetero-dimeric a308/c309 with an edge-to-edge distance of 3.7 Å (Fig. 2b). The axial ligands of the central Mg atom in a301, c302, a303, c304, c306, a307, and c309 are coordinated by side chains of E64, H67, H81, Q119, E128, E163, and N166, respectively, whereas those in a305, a308, and a310 are coordinated by water molecules of w332, w333, and w334, respectively (Fig. 2b). The pigment composition of Sm3-

Fcpb1 is the same as that of Sm1-Fcpb1 (Supplementary Fig. 5), whereas Sm4-Fcpb1 has a302 instead of c302, forming a hetero-dimeric Chl pair of a302/c306 (Supplementary Fig. 5). This indicates a different composition of pigment molecules even with the same Fcpb1 protein. It is not clear if the differences in the pigment compositions of the same protein have physiological meanings or are due to different locations of the proteins. It should be noted that the water molecules of w332 and w334 in Sm1-Fcpb1 are not observed in both Sm3-Fcpb1 and Sm4-Fcpb1, because of the very weak densities of the water molecules.

Fcpb2 has high sequence similarity to Fcpb1 (93%) (Supplementary Table 4); nevertheless, we identified Sm2 as Fcpb2 because of the difference in the density map between Fcpb2-F122 and Fcpb1-A122 (Supplementary Figs. 6, 7a, b, h). F122 has a much large side chain than A122, and can be seen as an indicator for the assignment of Fcpb2 (Supplementary Figs. 7b, h). Among the FCP sequences compared, Fcpb6 has F137 corresponding to the residue of Fcpb2-F122 (Supplementary Figs. 6, 7h). However, there are distinctive differences between Fcpb2-R110 and Fcpb6-F125 (Supplementary Fig. 7b, f, h) and between Fcpb2-F123/F124 and Fcpb6-A138/I139 (Supplementary Figs. 6, 7h). These differences excluded the assignment of Fcpb6 to Sm2.

Sm2-Fcpb2 has 3 Chls *c* and 7 Chls *a* in addition to 6 Fxs (Supplementary Table 3). Chl c302 in Sm1-Fcpb1 is changed to a302 in Sm2-Fcpb2, forming a hetero-dimeric a302/c306 pair (Fig. 2c). The axial ligands of 9 Chls except for a310 in Sm2-Fcpb2 are the same as those of 9 Chls in Sm1-Fcpb1. The ligand of a310 is a water molecule (w334) in Sm1-Fcpb1 (Fig. 2b), but this water molecule cannot be seen in Sm2-Fcpb2, likely due to its weak density.

Among Fcpb1s, Sm1 interacts with Sm4 through interactions between Sm1-K151 and Sm4-E155 and between Sm1-W131 and Sm4-a308 at distances of 3.5–3.7 Å at the stromal side (Supplementary Fig. 8a), whereas Sm1-Fx325 interacts with Sm4-Fx326 at a distance of 3.4 Å at the lumenal side (Supplementary Fig. 8b). Similar interactions are found between Sm2-W131 and Sm1-a308 (Supplementary Fig. 8c), Sm2-D152 and Sm3-K151, Sm3-W131 and Sm2-a308 (Supplementary Fig. 8d), at the stromal side, and between Sm2-Fx325 and Sm1-Fx326 (Supplementary Fig. 8e), Sm3-Fx325 and Sm2-Fx326 (Supplementary Fig. 8f), at the lumenal side, respectively. However, no salt bridge exists between Sm1 and Sm2 (Supplementary Fig. 8c), because Sm2-Fcpb2 has E151 instead of a lysine residue corresponding to K151 in Sm1-Fcpb1 (Supplementary Fig. 6). It should be noted that the NH group of W131 is hydrogen-bonded to a nitrogen atom of a308 but it is not a direct ligand of the Mg atom in a308.

**Structure of the M-tetramer**. The M-tetramer is composed of two Fcpb1, one Fcpb3, and one Fcpb4 (Fig. 1). Both Mm3 and Mm4 are identified as Fcpb1, whereas Mm1 and Mm2 are identified as Fcpb3 and Fcpb4, respectively. The sequence of Fcpb3 has a similarity of 69% with that of Fcpb1 (Supplementary Table 4). Characteristic residues of Fcpb3-F112/Q113 are significantly different from the corresponding residues of Fcpb1-R110/A111, respectively, contributing to the assignment of Mm1 as Fcpb3 (Supplementary Figs. 6, 7a, c, h). Mm1-Fcpb3 has 4 Chls *c* and 5 Chls *a* in addition to 5 Fxs and 1 Ddx (Fig. 3a and Supplementary Table 3). Compared with Sm1-Fcpb1, Mm1-Fcpb3 has Ddx323 instead of Fx, and lacks one Chl molecule (a305). Most of the ligands of Chls are conserved between Mm1-Fcpb3 and Sm1-Fcpb1; the side chains of E66, H69, H83, Q121, E130, E167, and N170 in Mm1-Fcpb3 are coordinated to the Mg atoms of a301, c302, a303, c304, c306, a307, and c309, respectively, whereas the water molecule w333 is coordinated to the Mg

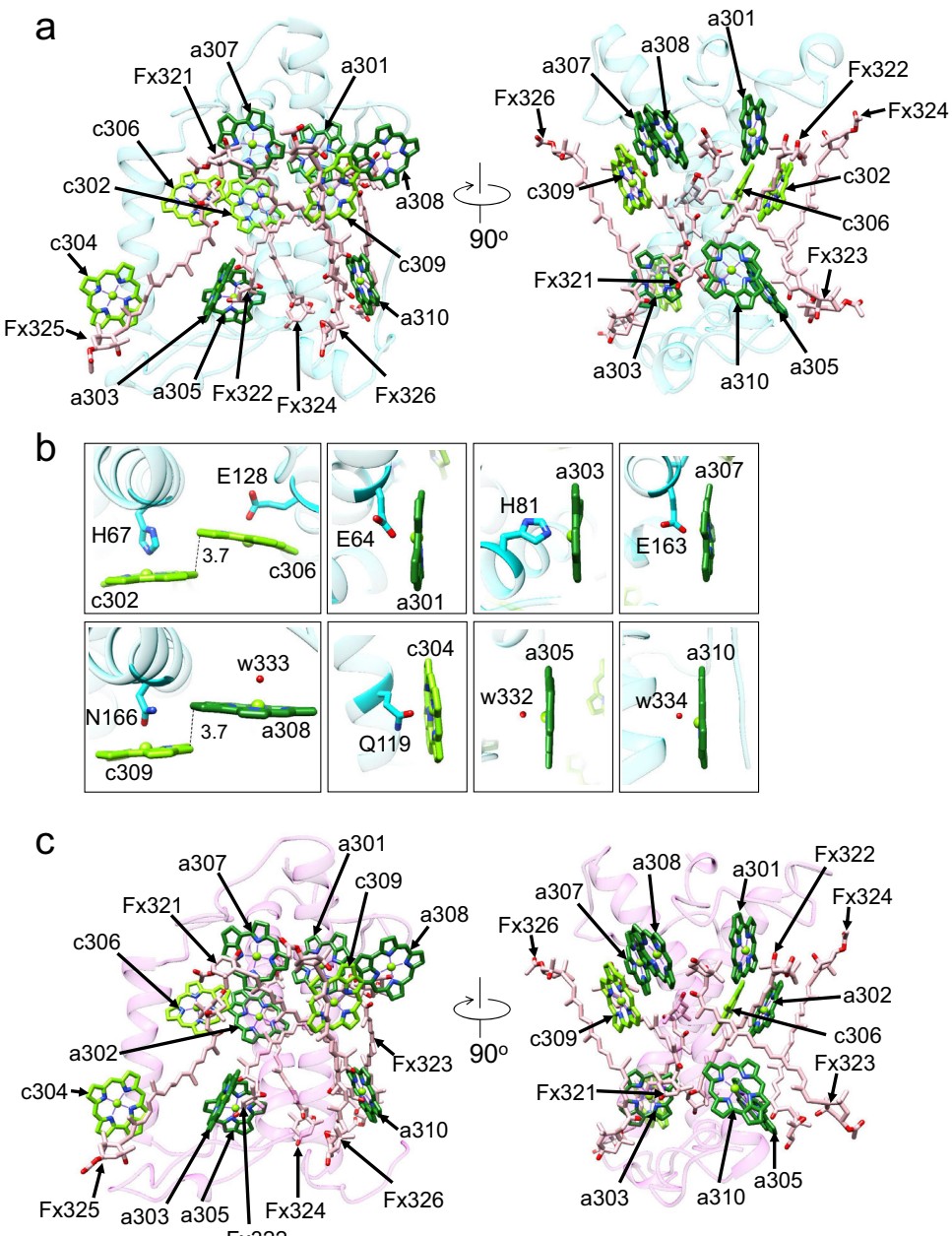

**Fig. 2 Structure of the characteristic subunits in the S-tetramer. a** Structure of Sm1-Fcpb1 depicted with Cα atoms and arrangement of the pigments (Chls and Cars), with the right-side panel rotated 90° clockwise relative to the left-side panel. The pigments of Chl *a*, Chl *c*, and Fx are colored "forest green", "bright green", and "pink", respectively. Only the rings of Chls are depicted. **b** Characteristic ligands to the Mg atom in each Chl molecule. Interactions are indicated by black dashed lines and numbers are distances in Å. **c** Structure of Sm2-Fcpb2 and arrangement of the pigments (Chls and Cars).

atom of a308. The ligand of the Mg atom in a310 of Mm1-Fcpb3 is changed to the side chain of H184 instead of a water molecule seen in Sm1-Fcpb1 (Fig. 3b).

Fcpb4 has high sequence similarity with Fcpb1 (92%) (Supplementary Table 4). However, we can distinguish Fcpb4 from Fcpb1 at the position of Mm2. This is due to a difference between Fcpb4-F110 and Fcpb1-R110 (Supplementary Figs. 6, 7a, d, h). In addition, Fcpb4-L124/G125/F126 are different from Fcpb1-F124/A125/L126 (Supplementary Figs. 6, 7h). The differences in these four amino acid residues lead to an assignment of Fcpb4 at the position of Mm2. Mm2-Fcpb4 has 4 Chls *c* and 5 Chls *a* in addition to 6 Fxs, and does not possess a305 compared with Sm1-Fcpb1 (Fig. 3c and Supplementary Table 3). The Mg

ligands for all 9 Chls are conserved between Mm2-Fcpb4 and Sm1-Fcpb1.

Mm3-Fcpb1 forms a hetero-dimeric a302/c306 different from Sm1-Fcpb1, and it does not possess Fx323 (Supplementary Fig. 5). In contrast, the pigment composition of Mm4-Fcpb1 is consistent with that of Sm1-Fcpb1 (Supplementary Fig. 5). This reinforces the different composition of pigment molecules even among the same Fcpb1 apoproteins. It should be noted that the two water molecules w332 and w334 observed in Sm1-Fcpb1 are not seen in both Mm3-Fcpb1 and Mm4-Fcpb1, because of the very weak densities of these water molecules.

Mm1 interacts with Mm2 between Mm1-Q156/K159 and Mm2-N148 and between Mm1-a308 and Mm2-F126/W131 at

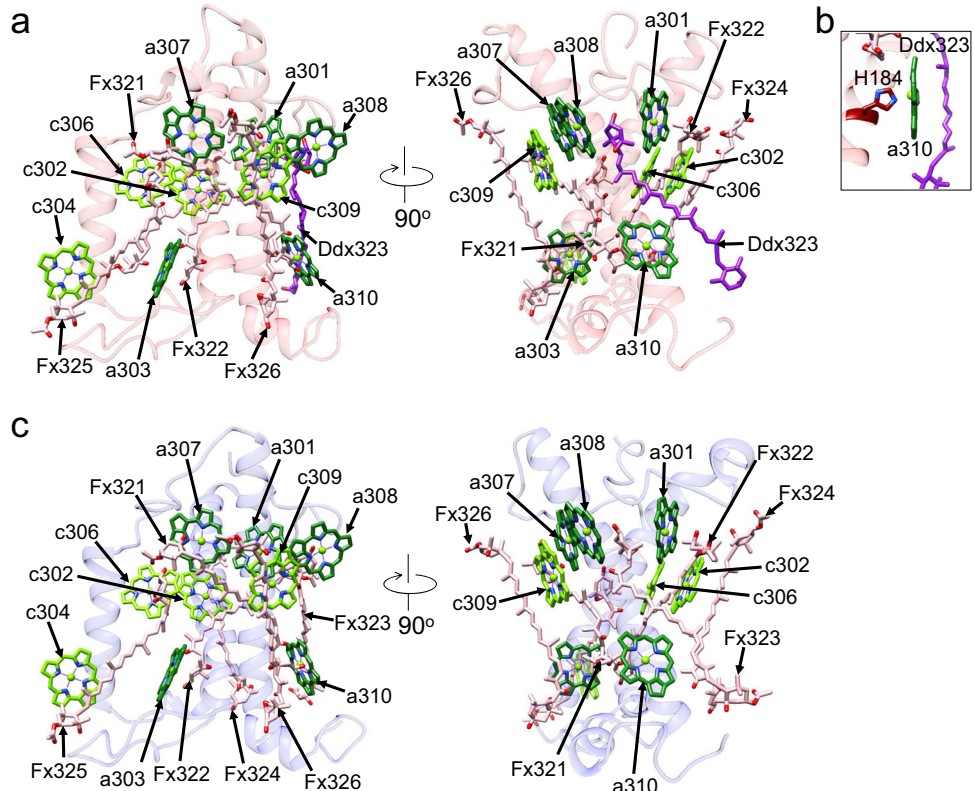

**Fig. 3 Structure of the characteristic subunits in the M-tetramer. a** Structure of Mm1-Fcpb3 depicted with Cα atoms and arrangement of the pigments (Chls and Cars), with the right-side panel rotated 90° clockwise relative to the left-side panel. The pigments of Chl *a*, Chl *c*, Fx, and Ddx are colored "forest green", "bright green", "pink", and "purple", respectively. Only rings of the Chls are depicted. **b** Amino-acid ligand to the Mg atom in a310. **c** Structure of Mm2-Fcpb4 and arrangement of the pigments (Chls and Cars).

distances of 2.5–3.5 Å at the stromal side (Supplementary Fig. 8g), and between Mm1-Fx326 and Mm2-Fx325 at the lumenal side (Supplementary Fig. 8h). Both Mm1-Fx326 and Mm2-Fx325 also interact with Mm1-SQDG329 at distances of 3.2–3.6 Å. Mm1 also interacts with Mm4 between Mm1-F128/W133 and Mm4-a308 at distances of 3.6–3.9 Å at the stromal side (Supplementary Fig. 8i), and between Mm1-Fx325 and Mm4-Fx326 at a distance of 3.4 Å at the lumenal side (Supplementary Fig. 8j). Mm2 interacts with Mm3 between Mm2-a308 and Mm3-W131 at distances of 3.2 Å at the stromal side (Supplementary Fig. 8k), and between Mm2-Fx326 and Mm3-Fx325 at a distance of 3.5 Å at the lumenal side (Supplementary Fig. 8i). As noted in the case of the S-tetramer, W131 is hydrogen-bonded to an N atom of a308 but not a direct ligand of the Mg atom of a308 in the M-tetramer.

**Structures of the monomeric FCP subunits.** The three FCPII monomers m1, m2, and m3 are identified as Fcpb5, Fcpb6, and Fcpb7, respectively. The sequence of Fcpb5 has a similarity of 33% with that of Fcpb1 (Supplementary Table 4). Characteristic residues of Fcpb5-S156/Q157 are remarkably different from the corresponding residues of Fcpb1-R110/A111, respectively, identifying m1 as Fcpb5 (Supplementary Figs. 6, 7a, e, h). The subunit m1-Fcpb5 binds 1 Chl *c* and 9 Chls *a* in addition to 2 Fxs and 1 Ddx (Fig. 4a and Supplementary Table 3). Compared with Sm1-Fcpb1, three Chls *c* are substituted by Chl *a* (a302/a304/a306), and one Chl molecule a305 is lacking in m1-Fcpb5. There is a homo-dimeric a302/a306 in m1-Fcpb5 in addition to the hetero-dimeric a308/c309; the latter hetero-dimer is the same as that seen in Sm1-Fcpb1 (Supplementary Fig. 9a). The axial ligands of the Mg atom in a301, a302, a304, a306, a307, c309, and a310 are

provided by side chains of E112, H115, Q165, E174, E221, H224, and Q238, respectively, whereas those of a303 and a308 are a water molecule w331 and a phosphatidylglycerol PG326, respectively (Supplementary Fig. 9a). No axial ligand exists in a311, likely due to the weak density of the water molecule. Moreover, one Fx molecule is substituted by Ddx322, and three Fx molecules Fx323, Fx324, and Fx326 are lacking.

The sequence of Fcpb6 has a similarity of 64% with that of Fcpb1 (Supplementary Table 4). Characteristic residues of Fcpb6-F125/F137 are different from the corresponding residues of Fcpb1-R110/A122, respectively, leading to an assignment of m2 as Fcpb6 (Supplementary Figs. 6, 7a, f, h). The subunit m2-Fcpb6 binds 3 Chls *c* and 6 Chls *a* in addition to 2 Fxs and 2 Ddxs (Fig. 4b and Supplementary Table 3). Compared with Sm1-Fcpb1, one Chl *c* is replaced by a309, and three Chls a305, a308, and a310 are lacking in m2-Fcpb6. Instead, two additional Chls (a312 and a313) exist in m2-Fcpb6. Therefore, m2-Fcpb6 has only one homo-dimeric c302/c306. The axial ligands of the Mg atom in a301, c302, a303, c304, c306, a307, a309, and a312 are provided by side chains of E79, H82, H96, Q134, E143, E181, Q184, and H69, respectively, whereas that of a313 is provided by the carbonyl oxygen atom of the main chain of G161 (Supplementary Fig. 9b). Two Fx molecules are substituted by Ddx322 and Ddx325, whereas two Fx molecules of Fx323 and Fx326 are lacking.

The sequence of Fcpb7 has a similarity of 56% with that of Fcpb1 (Supplementary Table 4). Characteristic residues of Fcpb7-S110/K111/F118 are significantly different from the corresponding residues of Fcpb1-R110/A111/A118, respectively, identifying m3 as Fcpb7 (Supplementary Figs. 6, 7a, g, h). The subunit m3-Fcpb7 has 2 Chls *c* and 7 Chls *a* in addition to 4 Fxs and 2 Ddxs

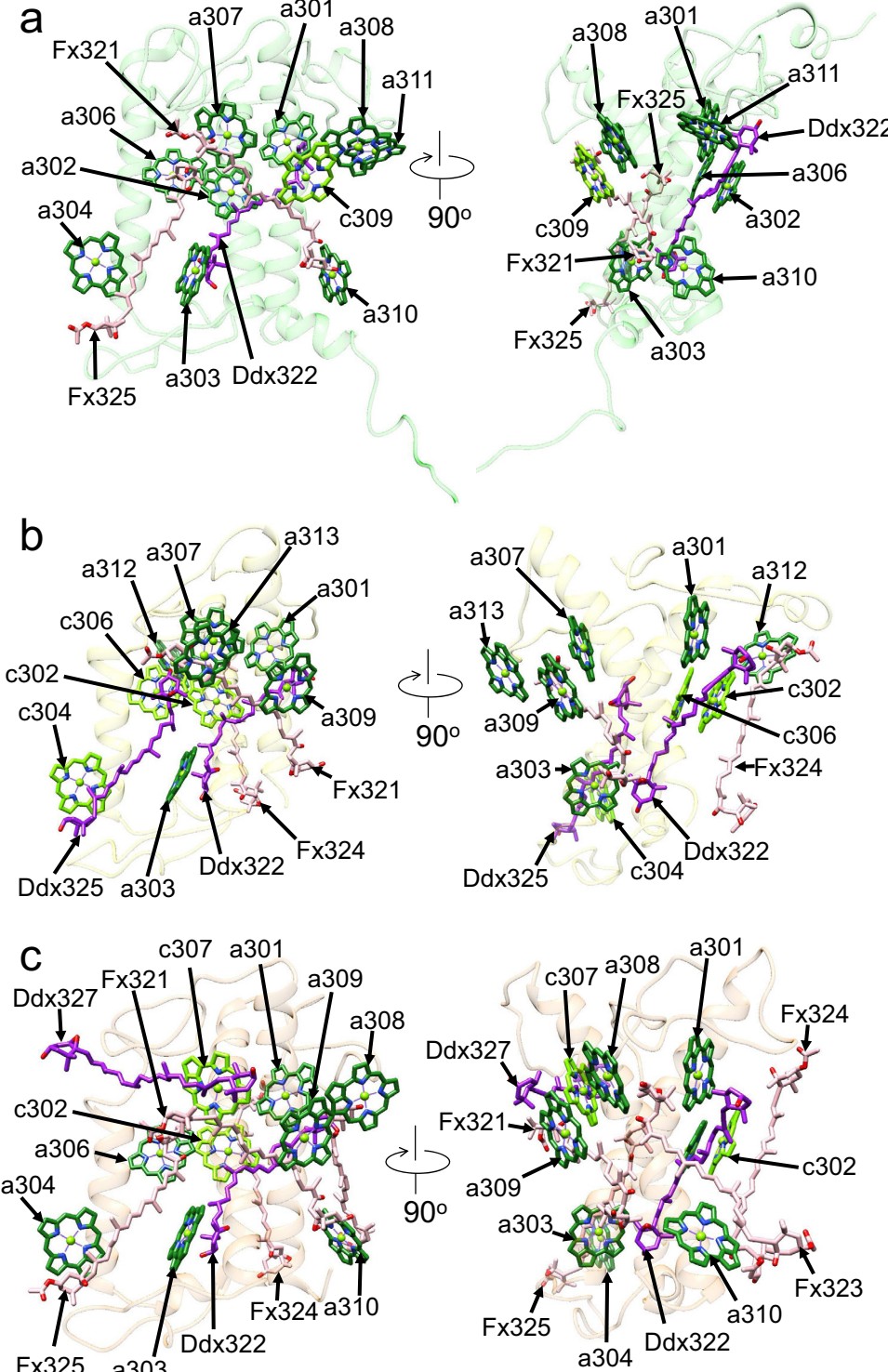

**Fig. 4 Structure of the three types of monomeric FCPs.** Structures of m1-Fcpb5 (**a**), m2-Fcpb6 (**b**), and m3-Fcpb7 (**c**) depicted with Cα atoms and arrangements of the pigments (Chls and Cars), with the right-side panel rotated 90° clockwise relative to the left-side panel. The pigments of Chl *a*, Chl *c*, Fx, and Ddx are colored "forest green", "bright green", "pink", and "purple", respectively. Only rings of the Chls are depicted.

(Fig. 4c and Supplementary Table 3). Compared with Sm1-Fcpb1, three Chls *c* are substituted to a304/a306/a309, and one Chl *a* is replaced by c307 in m3-Fcpb7. One Chl a305 is lacking in m3-Fcpb7. Both hetero-dimeric c302/a306 and homo-dimeric a308/a309 are found in m3-Fcpb7. One Fx is substituted to Ddx322, whereas Fx326 is lacking and Ddx327 is present. The axial ligands of the Mg atom in a301, c302, a303, a304, a306, c307, a309, and

a310 are provided by side chains of E65, H68, E82, Q119, E128, E168, Q171, and H185, respectively (Supplementary Fig. 9c). Since the ligand of a308 cannot be seen in the present structure, its plausible candidate may be a water molecule.

**Interactions among the FCPII units.** The m3 subunit is closely associated with the M-tetramer as reported previously[7,8]. In

particular, the phytol chain of Mm2-a310 is anchored to m3, and F206 in Mm2 interacts with F106 in m3 at a distance of 3.9 Å at the lumenal side (Supplementary Fig. 8m). In addition, Mm2-PG327 is anchored at the center of three subunits Mm2, Mm3, and m3, and has close interactions with m3-D114, Mm2-F197, and Mm3-S94 at distances of 3.0–4.0 Å. At the stromal side (Supplementary Fig. 8n), m3-PG328 is anchored at the interface between m3 and Mm2, with close interactions with Mm2-F44 and m3-Y63/K133 at distances of 3.5–4.1 Å. In addition, m3-PG328 is closely located near m3-a306 and Mm2-Fx323.

The m2 subunit interacts with Mm1 closely, as reported previously[7,8]. In particular, m2-T133 interacts with Mm1-F112 at a distance of 3.6 Å at the lumenal side (Supplementary Fig. 8o), and Mm1-Fx324 is located in the interface between m2 and Mm1 (Supplementary Fig. 8o). In addition, the interactions of m2 with Mm1 are likely strengthened by the insertions of both SQDG327 and MGDG328 in Mm1 (Supplementary Fig. 8o).

The m1 subunit is associated with the M-tetramer rather than the S-tetramer, as reported previously[7,8]. In particular, m1-S44/L81/F83/V86/V88 interact with Mm1-L44/F47/D48/L50 at distances of 3.4–4.5 Å, and the phytol chain of m1-a302 is anchored to Mm1 at the stromal side (Supplementary Fig. 8p). In addition, m1-I95/K182 are associated with Mm2-Q136/Y137 at distances of 2.4–3.3 Å (Supplementary Fig. 8q), and Mm2-PG327 is inserted into the interface between m1 and Mm2. At the lumenal side, Mm2-MGDG328 interacts with Mm2-S94 at a distance of 3.7 Å, and is anchored between m1 and Mm2 (Supplementary Fig. 8r). The m1-Q240 is associated with Mm1-S202 at a distance of 3.1 Å, whereas m1-a310 interacts with Mm1-F211 at a distance of 3.5 Å (Supplementary Fig. 8r).

**Excitation-energy transfer in the PSII-FCPII supercomplex**. To examine excitation-energy-transfer processes in the PSII-FCPII supercomplex, we measured time-resolved fluorescence (TRF) spectra at 77 K and obtained fluorescence decay-associated (FDA) spectra by global analysis. (Fig. 5). Five-lifetime components were necessary to fit the fluorescence rise and decay profiles globally. The first FDA spectrum (45 ps) exhibits a set of two positive bands at around 642 and 675 nm and a broad negative band at around 688 nm, reflecting energy transfer from Chls fluorescing at 642 and 675 nm to those at 688 nm. The second FDA spectrum (210 ps) shows a positive band at around 687 nm, while the third FDA spectrum (1.4 ns) depicts two positive bands at around 688 and 694 nm. The fourth FDA spectrum (5.4 ns) exhibits two positive bands at around 688 and 695 nm. The fifth FDA spectrum (32 ns) displays delayed fluorescence with positive bands at around 688 and 693 nm, suggesting charge recombination in PSII[20].

We have previously reported excitation-energy transfer in two types of diatom PSII preparations, PSII complexes and PSII-FCPII particles[21]. The former complex contains almost no FCPs, and the latter has a large amount of FCPs with lower purity than the PSII-FCPII supercomplex examined here. Combining with the previous results[21], the 642 and 675 nm bands observed in the first FDA spectrum here appear to originate from Chls c and Chls a in FCPII, respectively, whereas the broad band at 688 nm in the first FDA spectrum may be attributed to low-energy Chls in PSII and/or in the interface between PSII and FCPII. This suggests that energy transfer from FCPII to PSII occurs with a time constant of 45 ps. The 687 nm band with a tail at the shorter-wavelength side in the second FDA spectrum appears to transfer excitation energy to reaction-center (RC) Chls, followed by trapping, and hence it may originate from Chls in PSII. The 688 and 694 nm bands in the third FDA spectrum suggest energy trapping to low-energy Chls in CP43 and CP47, respectively, and they may be involved in

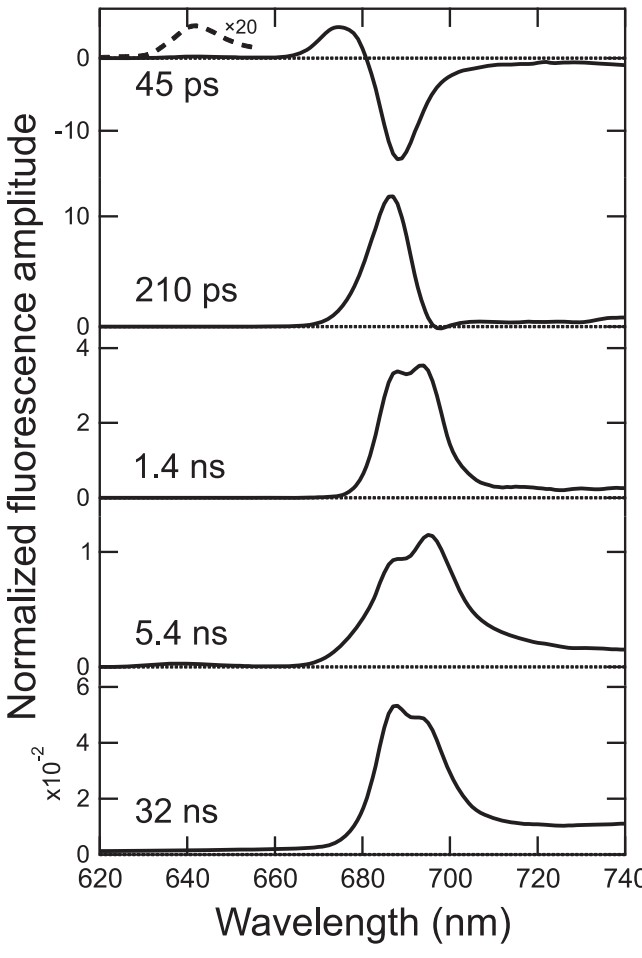

**Fig. 5 Fluorescence decay-associated spectra of the PSII-FCPII supercomplex at 77 K.** The spectra were normalized by the total fluorescence intensity. Dashed spectrum was obtained by multiplication of a factor of 20, in order to visualize the 642 nm decay component.

slow energy quenching. The 688 and 695 nm bands in the fourth FDA spectrum indicate energy termination in CP43 and CP47, respectively, because the time constant is comparable to a lifetime of the $S_1$ state of Chl a (~5.0 ns)[22,23].

## Discussion

The previous structures of PSII-FCPII showed the association of both S and M homo-tetramers with PSII, but the individual sequences of each protomer in the tetramers were not identified or had different sequences[7,8]. Our present refined structure has identified different types of FCPII-monomer units that form the S and M hetero-tetramers (Fig. 1), albeit with very similar amino-acid sequences among them (Fcpb1, Fcpb2, and Fcpb4) (Supplementary Fig. 6 and Supplementary Table 4). This is realized by both high-quality maps that allow us to identify side chains of characteristic amino acid residues in each FCP (Supplementary Fig. 7) and the information of the genome of C. gracilis[11]. As a result (Fig. 1), the S-tetramer is composed of three Fcpb1s (Sm1, Sm3, and Sm4) and one Fcpb2 (Sm2), whereas the M-tetramer is composed of two Fcpb1s (Mm3 and Mm4), one Fcpb3 (Mm1), and one Fcpb4 (Mm2).

Differences are also found in the pigment content and their binding patterns among these various FCPII-monomer units within the tetramers. Fcpb1s (Sm1, Sm3, Sm4, Mm3, and Mm4) show different pigment compositions; the ratio of

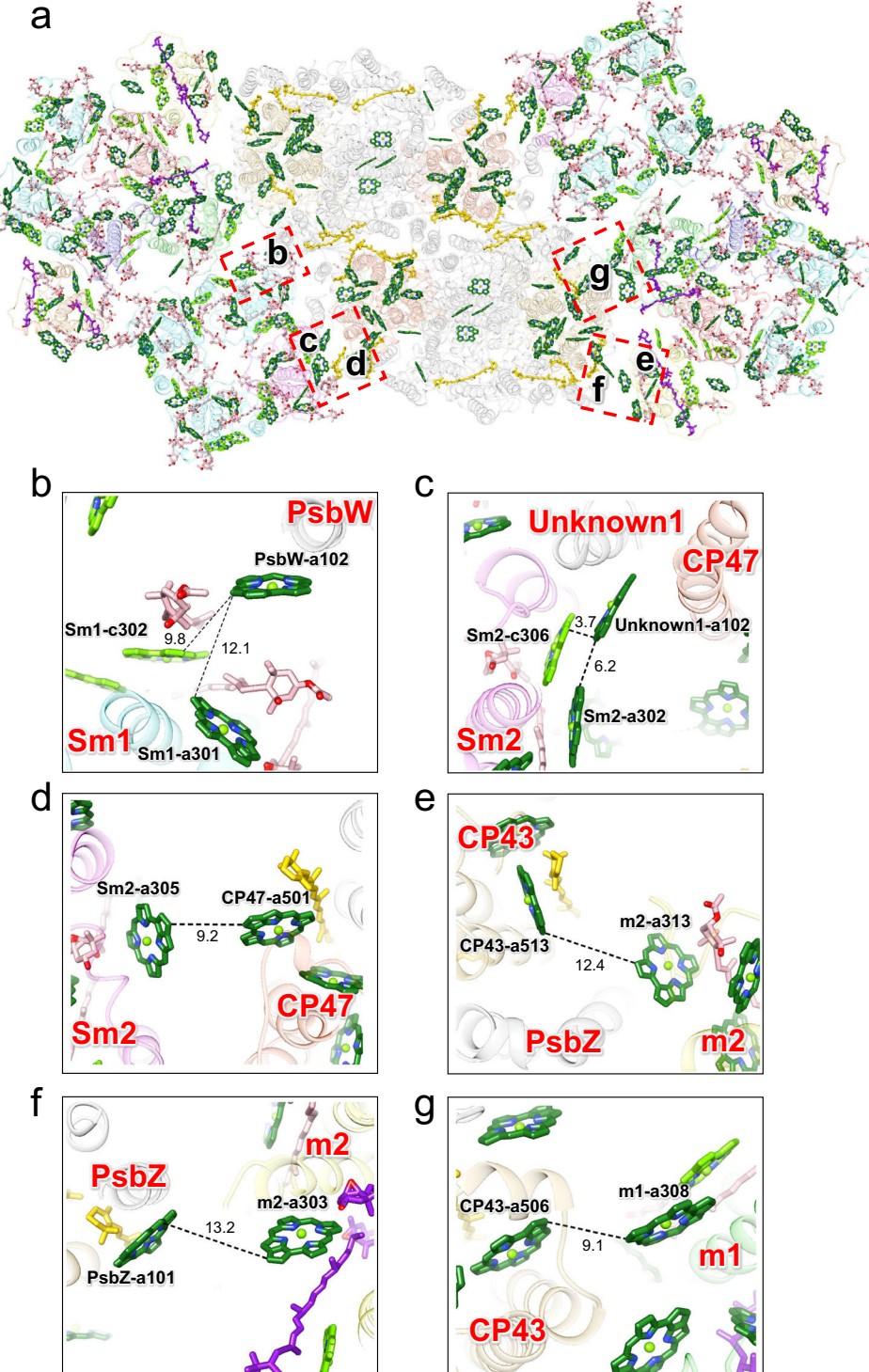

**Fig. 6 Excitation-energy-transfer pathways from FCPII to PSII. a** Pigment-pigment interactions in the PSII-FCPII. The structure is viewed from the stromal side. Colors of the FCPII subunits are the same as in Fig. 1. CP47 and CP43 are colored "orange red" and "goldenrod", respectively. The pigments of Chl *a*, Chl *c*, Fx, Ddx, and *β*-carotene are colored "forest green", "bright green", "pink", "purple", and "gold", respectively. Only rings of the Chls are depicted. Red squared areas are enlarged in panels **b**–**g**, in order to show energy-transfer pathways from FCPII to PSII. Interactions are indicated by dashed lines and numbers are distances in Å. **b** Interactions between Sm1 and PsbW from the adjacent PSII-FCPII unit viewed from the stromal side. **c** Interactions between Sm2 and CP47/Unknown1 viewed from the stromal side. **d** Interactions between Sm2 and CP47 viewed from the lumenal side. **e** Interactions between m2 and CP43/PsbZ viewed from the stromal side. **f** Interactions between m2 and PsbZ viewed from the lumenal side. **g** Interactions between m1 and CP43 viewed from the stromal side.

Chl $a$:Chl $c$ is 6:4 in Sm1/Sm3/Mm4 and 7:3 in Sm4/Mm3 (Supplementary Table 3). This was determined according to a threshold of 15 σ contour level in the cryo-EM map (Supplementary Fig. 4b; see "Methods" for details). The different pigment compositions among Fcpb1s might contribute to the characteristic excitation-energy transfer and assembly features of the FCPII units in vivo. However, it should be noted that the pigment assignment is dependent on the map quality. Further studies at a higher resolution would be required for verifying the heterogeneity of the pigment compositions among Fcpb1s located in different positions.

The three types of FCP monomers (m1, m2, and m3) in the previous study of Nagao et al. were assigned to polyalanines[7], whereas those in the other study of Pi et al. were assigned to sequence mixtures of polyalanines, transcriptome data of *C. gracilis*, and FCP sequences from other diatom species[8]. In the present study, we assigned these three monomers to Fcpb5, Fcpb6, and Fcpb7, respectively. The sequence of Fcpb5 is virtually identical to that of FCP-D reported by Pi et al.[8], with differences in only five amino-acid residues. The pigment ratio of m1 is 9:1:2:1 for Chl $a$:Chl $c$:Fx:Ddx assigned at a threshold of 15 σ contour level in the cryo-EM map (Supplementary Table 3), which is also identical to that reported by Pi et al.[8] (see Table S3 therein). In contrast, Pi et al. assigned the sequences of FCP-E and FCP-F (corresponding to m2 and m3, respectively) to polyalanines and Lhcf4 of the diatom *Phaeodactylum tricornutum* (see Table S2 in ref. [8]). These sequences are largely different from the Fcpb6 and Fcpb7 sequences assigned for m2 and m3, respectively, in the present study. The pigment ratio of m2 in this study is Chl $a$:Chl $c$:Fx:Ddx = 6:3:2:2 assigned at a threshold of 15 σ contour level (Supplementary Table 3), whereas that of FCP-E was Chl $a$:Chl $c$:Fx:Ddx = 9:2:6:0 (see Table S3 in ref. [8]). The pigment ratio of m3 in this study is Chl $a$:Chl $c$:Fx:Ddx = 7:2:4:2 (Supplementary Table 3), whereas that of FCP-F was Chl $a$:Chl $c$:Fx:Ddx = 7:2:6:0 (see Table S3 in ref. [8]). Thus, the refined PSII-FCPII structure reported here offers new findings for the subunit and pigment assignments of not only the S/M hetero-tetramers but also the three types of monomeric FCP subunits.

Interactions among the individual monomer units in the S and M hetero-tetramers resemble significantly at both lumenal and stromal sides (Supplementary Fig. 8a–l). However, the present refined structure of PSII-FCPII has unveiled remarkable differences between this study and the previous studies[7,8]. For example, one Fcpb2 is positioned at Sm2 in the S-tetramer (Fig. 1), and Sm2-F122 closely interacts with Unknown1 of the PSII core (Supplementary Fig. 10). Because F122 is replaced by alanine or leucine in the corresponding residue of Fcpb1, Fcpb3, and Fcpb4 (Supplementary Figs. 6, 7h), Sm2-F122 seems to play a critical role in the association of the S-tetramer with the PSII core. The characteristic phenylalanine corresponding to Fcpb2-F122 is absent in the M-tetramer; therefore, the M-tetramer is not associated with Unknown1. This results in no direct association of the M-tetramer with the PSII core (hence its name "moderately associated"). In contrast, Fcpb3 and Fcpb4 are positioned at Mm1 and Mm2, respectively (Fig. 1), and there are multiple interactions of Mm1 and Mm2 with m1, m2, and m3 (Supplementary Fig. 8m–r). Mm1 seems to be essential for the association with m1 and m2, whereas Mm2 can contribute to the binding of m3. These structural differences may be necessary for the proper association of each FCP tetramer to their respective positions during the assembly processes of FCPII.

This study improved the assignment of pigment molecules in the PSII-FCPII supercomplex remarkably compared with the previous studies[7,8]. Based on these assignments, we can more accurately discuss excitation-energy-transfer pathways in the PSII-FCPII supercomplex. In the following, we combine the results of FDA spectra (Fig. 5) and our previous TRF analyses[21] to decipher the complicated excitation-energy-transfer pathways in the supercomplex.

The first FDA spectrum of PSII-FCPII (Fig. 5) indicates energy transfer from FCPII to PSII. Based on the spectroscopic findings and the refined PSII-FCPII structure, we present energy-transfer pathways from FCPII to PSII (Fig. 6a). In the interfaces between FCPII and PSII, there are a large number of possible energy transfer pathways, which include pathways from Sm1-c302/a301 to PsbW-a102 of the adjacent PSII-FCPII (Fig. 6b), from Sm2-a302/a305/c306 to Unknown1-a102/CP47-a501 (Fig. 6c, d), from m2-a303/a313 to CP43-a513/PsbZ-a101 (Fig. 6e, f), and from m1-a308 to CP43-a506 (Fig. 6g). The three characteristic bands observed in the first FDA spectrum (two positive bands at around 642 and 675 nm and one negative band at around 688 nm) (Fig. 5) should represent the Chl components located at the interfaces between PSII and FCPII (Fig. 6b–g). Since the positive bands at 642 and 675 nm act as energy donors, they can be assigned to Chls in FCPII. The 642 nm band may originate from Sm1-c302 and Sm2-c306 based on their short wavelengths, whereas the 675 nm band may come from Sm1-a301, Sm2-a302, Sm2-a305, m1-a308, m2-a303, and m2-a313. In contrast, the negative band at 688-nm is energy acceptors, and can be assigned to Chls in PSII. Six Chl molecules, PsbW-a102, Unknown1-a102, PsbZ-a101, CP47-a501, CP43-a506, and CP43-a513, are potential candidates for the 688 nm negative band. The energy transfer from donor to acceptor occurs with the time constant of 45 ps (Fig. 5). Since the donor-acceptor distance between Sm1-c302 and PsbW-a102 is much larger than that between Sm2-c306 and Unknown1-a102, the former transfer pathway may be the most probable candidate for the 45 ps transfer. The latter pathway may occur within a few picoseconds as suggested by the femtosecond fluorescence up-conversion study[24].

Most of the energy after their transfer from FCPII to PSII is trapped on the RC Chls with the time constant of 210 ps. Slow energy quenching may occur with the time constant of 1.4 ns, and then energy termination takes place in the CP43 and CP47 Chls with the time constant of 5.4 ns, followed by delayed fluorescence in the CP43 and CP47 Chls with the time constant of 32 ns.

Our previous femtosecond fluorescence up-conversion study with FCPII isolated from the PSII-FCPII particles[24] has shown that energy transfer from Fxs/Chls $c$ to Chls $a$ occurs at hundreds of femtoseconds, while energy transfer among Chls $a$ takes place at a few picoseconds. The up-conversion study also suggested the presence of two types of Fxs and Chls $a$ with different energy levels. The sub-pico to picosecond energy transfer seems to occur among/within individual FCPII units, and subsequently the energy is trapped on Chls located at the interfaces between FCPII and PSII, which is then followed by transfer to PSII at 45 ps (Fig. 5). However, we cannot distinguish between low and high-energy forms of Chl $a$ or between those of Fx from the current PSII-FCPII structural data. Further study by employing quantum chemical calculations using this structure will be required to determine the energy levels of each pigment in FCPII.

Our previous FDA analysis with the PSII-FCPII particles has suggested the presence of two types of PSII-FCPII complexes: one is in the light-harvesting mode, and the other one is in the energy-quenching mode[21]. The quenching mode was represented by the appearance of the band at around 678 nm and two vibrational bands at around 711 and 733 nm in the second FDA spectrum of the PSII-FCPII particles (480 ps)[21]. In the present study, the second FDA spectrum of the PSII-FCPII supercomplex (210 ps) shows none of these bands (Fig. 5), suggesting that the PSII-FCPII structure reflects the light-harvesting mode only. Therefore, the presence or absence of the 678 nm decay appears

to be a good indicator for the presence or absence of PSII-FCPII in the quenching mode.

The existence of heterogeneous PSII-FCPII supercomplexes in *C. gracilis* is further supported by subunit compositions of FCPII. Three types of major FCP bands have been obtained from this diatom by SDS-PAGE analysis; they are named FCP-A, FCP-B, and FCP-C[25,26]. The PSII-FCPII particles contain all of the three types of FCP bands[12,26]. However, the PSII-FCPII supercomplexes purified from the PSII-FCPII particles are enriched in FCP-A only, and contain a small amount of FCP-B and FCP-C[7]. These observations suggest that FCP-A is the major component in the light-harvesting PSII-FCPII, whereas FCP-B and FCP-C may act as the energy-quenching component in PSII-FCPII.

In conclusion, our refined structure of the PSII-FCPII supercomplex from *C. gracilis* allows us to identify all of the FCPII subunits, which shows that the S- and M-tetramers are not homo-tetramers but hetero-tetramers with different subunit compositions. Differences are found in the pigment composition and binding patterns among the different FCPII subunits and even among the same FCPII subunit located in various positions, which may be necessary for the proper bindings and interactions of each tetramer with their neighboring pigment-protein complexes in the process of FCPII assembly, as well as for the proper energy-transfer processes to occur. Our TRF analysis together with previous biochemical analyses[7,12,21,25,26] suggests the presence of the light-harvesting and energy-quenching PSII-FCPII supercomplexes in this diatom. These heterogenous PSII-FCPII supercomplexes may be necessary for diatoms to adapt to light-limited specific growth environments under the water surface. The unique light-harvesting and quenching strategy in diatoms may contribute to the great success of diatoms in the aquatic environment.

## Methods

**Purification of the PSII-FCPII supercomplex from *C. gracilis*.** The marine centric diatom, *C. gracilis* (UTEX LB 2658), was grown in artificial seawater[12] at a photosynthetic photon flux density of 30 μmol photons m$^{-2}$ s$^{-1}$ at 30 °C with continuous bubbling of air containing 3% (v/v) $CO_2$[7,27,28]. Preparation and biochemical characterization of the PSII-FCPII supercomplex were carried out in our previous study[7].

**TRF spectroscopy and global analysis.** TRF spectra were recorded by a time-correlated single-photon counting system with a wavelength interval of 1 nm and a time interval of 2.44 or 24.42 ps[29]. The excitation source was a picosecond pulse diode laser (PiL047X; Advanced Laser Diode Systems) operated at 459 nm with a repetition rate of 3 MHz (37 pJ/pulse). The laser spot at the sample point was a rectangle with a size of 2 × 3 mm; therefore, the excitation-light intensity was less than $1.4 \times 10^9$ photons/(pulse cm$^2$). This value should be low enough to ignore the annihilation effect[30]. The details of the TRF-measurement conditions were described[31].

FDA spectra were constructed by global analysis[32]. The quality of the TRF measurements and the validity of fitting analysis were confirmed in Supplementary Fig. 11. Fluorescence rise and decay components are depicted in positive and negative peaks, respectively. A pair of positive and negative amplitudes reflect excitation-energy transfer from a pigment with a positive amplitude to one with a negative magnitude[33,34]. In the case of a sequential energy transfer of A→B→C (e.g., A, B, and C represent pigments), clear positive and negative bands appear only for A and C, respectively, whereas the negative amplitude of B (A→B transfer) is canceled by the positive one of B (B→C transfer) in the same FDA spectrum.

**Cryo-EM data collection.** Samples of the PSII-FCPII supercomplex (128 or 256 μg of Chl mL$^{-1}$) were embedded in vitreous ice and examined with a 300 kV cryo-electron microscope (Titan Krios, Thermo Fischer Scientific) equipped with a field emission gun, a Cs corrector (CEOS GmbH), and a direct electron detection camera (Falcon 3EC, Thermo Fischer Scientific). Briefly, automated data collection was performed by the EPU software (Thermo Fisher Scientific). Electron micrographic movies were recorded at a nominal magnification of ×59,000 using the Falcon 3EC detector in a linear mode. The movies were recorded with a pixel size of 1.113 Å and a dose rate of 20 electrons Å$^{-2}$ s$^{-1}$. The nominal defocus range was −1.5 to −3.5 μm. Each exposure of 2.5 s was dose-fractionated into 33 movie frames. In addition to the movies that we had analyzed (12,739 micrographs)[7], we acquired additional 8060 micrographs. All micrographs (20,799 micrographs in

total) were combined and used for the high-resolution structural analysis of the PSII-FCPII supercomplex.

**Cryo-EM image processing.** The movie frames were aligned and summed using the MotionCor2 software version 1.2.3[35] to obtain a final dose-weighted image. The contrast transfer function (CTF) was estimated with the CTFFIND4 program version 1.13[36]. All of the following processes were performed using RELION-3.0[37]. As described in Supplementary Fig. 1, a total of 8,093,924 particles were automatically picked from 20,799 micrographs and then used for reference-free 2D classification. The resultant 3,489,210 particles from the good 2D classes were subjected to first 3D classification with a C1 symmetry using an initial model of the diatom PSII-FCPII supercomplex (EMD-9777) after passing through a 60 Å low-pass filter. The resultant 1,060,488 particles from the good 3D classes were subjected to 3D structural refinement and post-processing, and then to CTF refinement and Bayesian polishing. The polished particles were employed for further 3D reconstruction for the overall structure of PSII-FCPII ($C_2S_2M_2$) and the peripheral structure of FCPII ($S_1M_1$).

For the overall PSII-FCPII ($C_2S_2M_2$), 3D-focused classification was applied to the 1,060,488 particles with a mask covering the $S_2M_2$ region with a C1 symmetry. Two classes (class 1 and class 2) were selected as good classes, in which 154,121 particles in class 1 were subjected to additional 3D-focused classification with a mask covering the $S_2M_2$ region with a C2 symmetry, which resulted in a good class (class III). On the other hand, 211,273 particles in class 2 were subjected to additional 3D-focused classification with a mask covering the $S_2M_2$ region with a C2 symmetry, which yielded good classes (class A and class B). A total of 210,825 particles in class III, class A, and class B were used for 3D structural refinement with a C2 symmetry and post-processing, and then to CTF refinement and final post-processing. The overall PSII-FCPII ($C_2S_2M_2$) structure was reconstructed at 2.5 Å resolution based on the gold-standard FSC with a cutoff value of 0.143[38]. The local resolution of the final map was calculated using RELION.

For structural analysis of the peripheral FCPII ($S_1M_1$), orientations of the 1,060,488 particles were expanded with a C2 symmetry. The expanded particles were subjected to 3D-focused classification with a mask covering the $S_1M_1$ region with a C1 symmetry. The resultant good class (class a; 456,913 particles) was used for 3D structural refinement and post-processing, and then to CTF refinement and Bayesian polishing. To refine the peripheral FCPII ($S_1M_1$) region, the PSII-core map was subtracted from the map of the good class (456,913 particles). The subtracted particles were subjected to 3D-focused classification with a mask covering the $M_1$ region with a C1 symmetry. In total 373,897 particles were obtained as the best class (class ii), which were used for the final 3D structural refinement with a C1 symmetry and post-processing. The peripheral FCPII ($S_1M_1$) structure was reconstructed at 2.8 Å resolution based on the gold-standard FSC with a cutoff value of 0.143. The local resolution of the final map was calculated using RELION.

**Model building and refinement.** The 2.8 Å peripheral FCPII map was used for the model building of FCPII. For the model building of FCPII subunits, the cryo-EM structure of *C. gracilis* FCPII (PDB codes: 6J40) was manually fitted into the 2.8 Å cryo-EM map using UCSF Chimera version 1.14[39], and then inspected and adjusted individually with Coot version 0.7.2[40]. All Fcpb subunits were assigned and built based on interpretable features from the density map with a threshold of 15 σ contour level, including regions enriched in bulky residues and axial ligands of Chls (Supplementary Fig. 7). Chls *a* and *c* were distinguished by inspection of the density map corresponding to the phytol chain with a threshold of 15 σ contour level, which was found to be the least level not to link the map of Chls with that of noise. All Chls *c* were identified as Chl *c*1, because of difficulties in the rigid distinction between Chl *c*1 and Chl *c*2 at the present resolution. For the assignment of Cars, Fx and Ddx were distinguished based on the density covering the head group of Cars with a threshold of 15 σ contour level as described above. Lipids were assigned based on the density covering the head group of lipids with a threshold of 15 σ contour level. The peripheral FCPII structure was refined with PHENIX version 1.14 (phenix.real_space_refine)[41] with geometric restraints for the protein–cofactor coordination. The final model was further validated with Mol-Probity version 4.4[42], EMRinger version 1.0.0[43], and *Q*-score[44].

The 2.5 Å PSII-FCPII map was used for the model building of the overall PSII-FCPII supercomplex. For the model building of PSII, the cryo-EM structure of *C. gracilis* PSII (PDB codes: 6J40) was first manually fitted into the 2.5 Å cryo-EM map with UCSF Chimera, and then manually adjusted with Coot. Finally, the model for the PSII-FCPII supercomplex was assembled by fitting the peripheral FCPII model to the 2.5 Å map with UCSF Chimera. The complete PSII-FCPII supercomplex structure was then refined in a similar manner as the peripheral FCPII. The final model was further validated with MolProbity, EMRinger, and *Q*-score. The statistics for all data collection and structure refinement were summarized in Supplementary Tables 1, 2. All structural figures were made by Chimera and PyMOL version 2.3.0[45].

Since the numbering of Chls and Cars in this paper were different from those of the PDB data, we listed the relationship of the pigment numbering in this paper with those in the PDB data in Supplementary Tables 5, 6.

**Statistics and reproducibility**. As described in the Methods section, numerous PSII-FCPII particles were picked up from the cryo-EM images and used for structural analysis with standard protocols. The data statistics and evaluation of the resolution are documented in Supplementary Figs. 1, 2 and Supplementary Table 1.

**Reporting summary**. Further information on research design is available in the Nature Research Reporting Summary linked to this article.

## Data availability

The data that support this study are available from the corresponding authors upon reasonable request. Atomic coordinates and cryo-EM maps for the reported structure of the overall PSII-FCPII and peripheral FCPII have been deposited in the Protein Data Bank under accession codes 7VD5 and 7VD6, respectively, and in the Electron Microscopy Data Bank under accession codes EMD-31905 and EMD-31906, respectively.

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

## Acknowledgements

This work was supported by the JSPS KAKENHI grant Nos. JP20K06528, JP21K19085 (R.N.), JP20H02914 (K.K.), JP20H031160 (K.I.), JP20H03194 (F.A.), JP16H06553 (S.A.), JP17H06433 (J.-R.S.).

## Author contributions

R.N., and J.-R.S. conceived the project; R.N. purified the PSII-FCPII supercomplex; T.S., and N.D. performed mass spectrometry analysis; K.I., and M.K. refined genome information of *C. gracilis* and provided the amino-acid sequences for structural

modeling; N.M. collected cryo-EM images; R.N. processed the cryo-EM data and reconstructed the final EM maps; R.N., K.K., and F.A. built the structural models; K.K. refined the final models; S.A. measured TRF spectra; M.Y. performed global analysis of the TRF spectra; R.N., M.Y., and S.A. proposed structural interpretation on the basis of spectroscopic analysis; R.N., and J.-R.S. wrote the manuscript, and all of the authors contributed to the interpretations of the results and improvement of the manuscript.

## Competing interests

Authors declare no competing interests.
