## [Peer Review File · Nature Communications]

Structural basis for different types of hetero-tetrameric light-harvesting complexes in a diatom PSII-FCPII supercomplexReviewers' Comments:

Reviewer #1:

Remarks to the Author:

This manuscript by Nagao et al. presents a cryo-EM structure of PSII-FCPII supercomplex at 2.5 angstrom resolution. Although two structures of the same supercomplex from the same source were reported before, the present work extends the earlier low resolutions to a higher resolution, which allowed the identification of different Fcp gene products in the supercomplex. They found that Lhcf1 is the major component in the two FCPII tetramers, while gene products of Lhcf5, Lhcf6, and Lhcf7 occupy the positions of Sm2, Mm1, and Mm2, respectively, and the three monomeric FCPs (m1, m2 and m3) are gene products of Lhcr17, Lhcf4, and Lhcf13, respectively. Moreover, they were able to identify the positions of Chls c and diadinoxanthins in the supercomplex, which is important to build the complicated pigment network and to investigate the quenching sites in the supercomplex. The high-resolution structural information thus is critical for understanding the complex assembly and energy transfer processes.

Specific Comments:

In this work, the authors improve the resolution of the supercomplex structure from 3.8 to 2.5 angstrom. Could the authors elaborate on what they did to improve the map quality. It was mentioned that they acquired additional 8,060 micrographs, is that all?

The authors assigned Sm2 as Fcpb2 because it contains a F122 instead of A122 in other isoforms. However, based on Supplementary figure 6, Fcpb6 also possess a phenylalanine in the corresponding position. How did they exclude Fcpb6?

In the manuscript, the authors mentioned that they assign FCPII subunits and pigments on the basis of interpretable features from the density map. However, they did not show these density maps. Since the identification of different Fcp isoforms is an important part of this work, they are strongly encouraged to show these evidences. It will be better that they provide density maps in Supplementary figures and describe in detail how they interpret these density maps in Method.

In this work, multiple Fcpb1 subunits were found to have different pigment composition. It is a bit confusing for this reviewer. It is reasonable that in photosynthetic organisms, some different protein isoforms may have the same pigment arrangement, and the same protein may change its pigment composition under different conditions. However, it is a little difficult to imagine that the same protein has different pigment composition in one supercomplex. Does this difference have physiological significance? On the other hand, to distinguish between Chl a and Chl c requires highly accurate density map, since the only difference of Chl a compared with Chl c is the presence of the phytol chain, which can be highly mobile and show no density at all. Therefore, it is possible that some Chls a were wrongly assigned as Chl c due to the poor density of their phytol tail. This possibility should be considered in discussion.

Page 7, in the middle, "However, we can distinguish Fcpb3 from Fcpb1 at the position of Mm2", Fcpb3 -> Fcpb4.

Page 10, Line 5, sixth -> fifth

Reviewer #2:

Remarks to the Author:

The manuscript entitled "Structural basis for different types of hetero-tetrameric light-harvesting complexes in a diatom PSII-FCPII supercomplex" by Nagao et al. describes the structure of the PSII-

FCPII supercomplex from the diatom *Chaetoceros gracilis* with a resolution of 2.5 Å. The authors were able to assign the FCP proteins of the S- and M- tetramer, as well as of the monomers m1, m2 and m3 to specific gene products. They also indicated that they identified the position of chlorophyll c and diadinoxanthin molecules in the structure. Additionally, they analyzed the excitation-energy transfer from FCPII to PSII by time-resolved fluorescence spectroscopy.

The study is technically sound, the presented work is in principle interesting, and the conclusions are justified by the data.

However, based on how the manuscript is written, it is not clear for me, which findings are actually new, and which have been published already in previous work. Dr Shen is corresponding author of two previous publications, which were published almost at the same time in *Science* (10.1126/science.aax4406) and *Nature Plants* (10.1038/s41477-019-0477-x), presenting the CryoEM structure of a PSII-FCPII complex from *C. gracilis*.

In the current manuscript, the authors state on page three:

“One of our previous PSII-FCPII structure was reported at a 3.8-Å resolution⁷, which could not assign the three types of monomeric FCP subunits as well as the characteristic pigment molecules of Chl c and diadinoxanthin (Ddx). On the other hand, the other report showed the PSII-FCPII structure at a resolution of 3.0 Å⁸; however, not all of the FCPII subunits were assigned because of the significantly lower resolution in the peripheral regions.”

referring to the two previous publications. I agree with the first statement, however the second statement refers to the *Science* paper and, in this work, the authors have assigned already the sequences of the S- and M- tetramers, of the three types of monomeric FCP subunits, as well as Chl c and diadinoxanthin in the structure. Therefore, I think the statement is misleading. I have no doubts, that the higher resolution of the presented structure is beneficial and enables clearer assignments of structural features with higher coverage. However, there are still unassigned parts in the structure and the authors should point out more clearly the actual new findings. For example, are the structural details presented in Fig.2 – 4 (especially Fig. 2b) only visible in the new structure or are they already visible in the old work (PDB: 6JLU)? Also energy transfer pathways have been discussed already in Pi et al. 2019. Are there substantial differences based on the improved resolution of the new structure or are the suggested pathways in principle the same? The authors should discuss previous results and ideas more thoroughly, also to get more lively results and discussion parts, which are currently rather schematic and difficult to read.

Reviewer #1:

This manuscript by Nagao et al. presents a cryo-EM structure of PSII-FCPII supercomplex at 2.5 angstrom resolution. Although two structures of the same supercomplex from the same source were reported before, the present work extends the earlier low resolutions to a higher resolution, which allowed the identification of different Fcp gene products in the supercomplex. They found that Lhcf1 is the major component in the two FCPII tetramers, while gene products of Lhcf5, Lhcf6, and Lhcf7 occupy the positions of Sm2, Mm1, and Mm2, respectively, and the three monomeric FCPs (m1, m2 and m3) are gene products of Lhcr17, Lhcf4, and Lhcf13, respectively. Moreover, they were able to identify the positions of Chls c and diadinoxanthins in the supercomplex, which is important to build the complicated pigment network and to investigate the quenching sites in the supercomplex. The high-resolution structural information thus is critical for understanding the complex assembly and energy transfer processes.

First of all, we thank the reviewer for his/her highly positive evaluation as well as important comments and suggestions to improve our manuscript.

Comment 1:

In this work, the authors improve the resolution of the supercomplex structure from 3.8 to 2.5 angstrom. Could the authors elaborate on what they did to improve the map quality. It was mentioned that they acquired additional 8,060 micrographs, is that all?

Author reply 1:

As described in the Method section of “Cryo-EM data collection.”, we used 20,613 micrographs to obtain the improved map. Of the 20,613 micrographs, 12,739 micrographs had been already obtained in our previous study [Nagao et al., 2019 Nat. Plants], while additional 8,060 micrographs were newly obtained in this study. In addition, we used Relion-3.0 in this study instead of Relion-2.0 in the previous study. These are the major factors that contributed to improve the resolution, and we revised the manuscript to make these clearer (lines 14-16, page 24).

Comment 2:

The authors assigned Sm2 as Fcpb2 because it contains a F122 instead of A122 in other isoforms. However, based on Supplementary figure 6, Fcpb6 also possess a

phenylalanine in the corresponding position. How did they exclude Fcpb6?

Author reply 2:

In addition to F122, the residues in Fcpb2 at positions 123/124 are F123/F124, whereas the corresponding positions of Fcpb6 are A138/I139. Furthermore, a residue 110 of Fcpb2 are R110, whereas the corresponding position of Fcpb6 are F125. Based on the cryo-EM map, the residues of Fcpb2 at these positions better suit with the map. Therefore, we excluded Fcpb6 as a candidate for Sm2. A figure was added (Supplementary Fig. 7) to show the fit of the residues with the cryo-EM map at these positions, and explanations were also added to the section of “Structure of the S-tetramer” to indicate these (lines 7-11, page 8).

Comment 3:

In the manuscript, the authors mentioned that they assign FCPII subunits and pigments on the basis of interpretable features from the density map. However, they did not show these density maps. Since the identification of different Fcp isoforms is an important part of this work, they are strongly encouraged to show these evidences. It will be better that they provide density maps in Supplementary figures and describe in detail how they interpret these density maps in Method.

Author reply 3:

According to the comments of the reviewer, we added the explanation of subunit assignments for Fcpb1-7 to the individual Result sections, i.e., Fcpb3 (lines 10-13, page 9), Fcpb4 (line 23 in page 9 to line 4 in page 10), Fcpb5 (lines 7-9, page 11), Fcpb6 (lines 22-24, page 11), and Fcpb7 (lines 11-13, page 12). We also added Supplementary Fig. 7 to show the identification of different Fcpb isoforms. Moreover, we modified the description regarding how we interpret the density maps in the Method section of “Model building and refinement” in the revised manuscript.

Comment 4:

In this work, multiple Fcpb1 subunits were found to have different pigment composition. It is a bit confusing for this reviewer. It is reasonable that in photosynthetic organisms, some different protein isoforms may have the same pigment arrangement, and the same protein may change its pigment composition under different conditions. However, it is a little difficult to imagine that the same protein has different pigment composition in one supercomplex. Does this difference have physiological significance? On the other hand,

to distinguish between Chl *a* and Chl *c* requires highly accurate density map, since the only difference of Chl *a* compared with Chl *c* is the presence of the phytol chain, which can be highly mobile and show no density at all. Therefore, it is possible that some Chls *a* were wrongly assigned as Chl *c* due to the poor density of their phytol tail. This possibility should be considered in discussion.

Author reply 4:

We identified pigment compositions of each FCP subunit according to a threshold of 15 σ contour level using Coot. This criterion resulted in different pigment compositions among Fcpb1s. As already described in the Method section, Chls *a* and *c* were distinguished by inspection of the density map corresponding to the phytol chain with a threshold of 15 σ contour level, which was found to be the least level not to link the map of Chls with that of noise.

We partially agree with the comments of the reviewer regarding the possibility of wrong assignment of pigments. This is because the pigment assignment depends on the map quality. Further study at higher resolutions will be required for verifying the heterogeneity of pigment compositions among Fcpb1s. To explain this, we added a paragraph to the Discussion section of “Identification of the FCP hetero-tetramers and their pigment arrangements” in the revised manuscript (line 12 in page 15 to line 10 in page 16). At present, we are not sure if the different pigment compositions of the same protein have physiological significance or not.

Comment 5:

Page 7, in the middle, “However, we can distinguish Fcpb3 from Fcpb1 at the position of Mm2”, Fcpb3 -> Fcpb4.

Author reply 5:

We modified it; thank you.

Comment 6:

Page 10, Line 5, sixth -> fifth

Author reply 6:

We modified it; thank you.

Reviewer #2

The manuscript entitled “Structural basis for different types of hetero-tetrameric light-harvesting complexes in a diatom PSII-FCPII supercomplex” by Nagao et al. describes the structure of the PSII-FCPII supercomplex from the diatom *Chaetoceros gracilis* with a resolution of 2.5 Å. The authors were able to assign the FCP proteins of the S- and M- tetramer, as well as of the monomers m1, m2 and m3 to specific gene products. They also indicated that they identified the position of chlorophyll c and diadinoxanthin molecules in the structure. Additionally, they analyzed the excitation-energy transfer from FCPII to PSII by time-resolved fluorescence spectroscopy.

The study is technically sound, the presented work is in principle interesting, and the conclusions are justified by the data.

First of all, we thank the reviewer for his/her highly positive evaluation as well as important comments and suggestions to improve our manuscript.

Comment:

However, based on how the manuscript is written, it is not clear for me, which findings are actually new, and which have been published already in previous work. Dr Shen is corresponding author of two previous publications, which were published almost at the same time in *Science* (10.1126/science.aax4406) and *Nature Plants* (10.1038/s41477-019-0477-x), presenting the CryoEM structure of a PSII-FCPII complex from *C. gracilis*.

In the current manuscript, the authors state on page three:

“One of our previous PSII-FCPII structure was reported at a 3.8-Å resolution⁷, which could not assign the three types of monomeric FCP subunits as well as the characteristic pigment molecules of Chl c and diadinoxanthin (Ddx). On the other hand, the other report showed the PSII-FCPII structure at a resolution of 3.0 Å⁸; however, not all of the FCPII subunits were assigned because of the significantly lower resolution in the peripheral regions.” referring to the two previous publications. I agree with the first statement, however the second statement refers to the *Science* paper and, in this work, the authors have assigned already the sequences of the S- and M- tetramers, of the three

types of monomeric FCP subunits, as well as Chl c and diadinoxanthin in the structure. Therefore, I think the statement is misleading. I have no doubts, that the higher resolution of the presented structure is beneficial and enables clearer assignments of structural features with higher coverage. However, there are still unassigned parts in the structure and the authors should point out more clearly the actual new findings. For example, are the structural details presented in Fig.2 – 4 (especially Fig. 2b) only visible in the new structure or are they already visible in the old work (PDB: 6JLU)? Also energy transfer pathways have been discussed already in Pi et al. 2019. Are there substantial differences based on the improved resolution of the new structure or are the suggested pathways in principle the same? The authors should discuss previous results and ideas more thoroughly, also to get more lively results and discussion parts, which are currently rather schematic and difficult to read.

Author reply:

The findings of this study are totally new compared with two previous publications. The details are described below.

[Assignments of subunits and pigments]

As for subunit assignments, in the previous 3.0-Å structure (Science paper), most of the sequences of FCPII subunits were not assigned. The Science paper showed assignment of FCPII subunits as chimera proteins, such as sequence mixtures of polyalanine, transcriptome data of *C. gracilis*, and other diatom species (see details for Table S2 in the Science paper). Among the FCPII subunits, two types of FCP tetramers were assigned as a single-gene product named as FCP-A in the Science paper; however, FCP-A was not consistent with Fcpb1 in this study and the Nature Plants paper, with a low-sequence similarity (45%). The subunits corresponding to m2 and m3 were modelled using Lhcf4 of a diatom *Phaeodactylum tricornutum*; therefore, it is really difficult to compare the corresponding sequences between this study and the Science paper. In contrast, the subunit named as FCP-D in the Science paper, corresponding to m1 in this study, was virtually identical to Fcpb5 in this study, with differences in only five amino acids.

As for pigment assignments, here we assigned all pigments with a threshold of 15 σ contour level (see Methods details). However, in the Science paper, there is no description about criterion of pigment assignments. Since the Science paper showed low-resolution map in the region of FCPII with a resolution of 3.5-4.5 Å, distinguishing Chl c from Chl a and Ddx from Fx seems to be difficult based on the criterion of the

present study and the Nature Plants paper. However, the pigment composition of m1 was consistent with that of FCP-D, which has a relatively good map quality among the FCPII subunits in the Science paper.

Based on these observations, the FCPII subunits other than m1 were newly identified in this study. To explain these contents, we added two new sections of “Identification of the FCP hetero-tetramers and their pigment arrangements” (line 12 in page 15 to line 10 in page 16) and “Identification of m1, m2, and m3 and their pigment arrangements” (line 12 in page 16 to line 7 in page 17) to the Discussion section in the revised manuscript.

The revised manuscript leaves all Figures 2-4 including Figure 4a, because the present assignments of gene names and subunits are necessary for understanding the complicated pigment network of PSII-FCPII for the readers.

[Energy-transfer pathway]

In this study, we improved pigment assignments compared with Nagao et al., 2019 Nature Plants and Pi et al., 2019 Science. Therefore, this study showed novel pigment networks, which have been mostly described in the Discussion section “Energy-transfer pathways in the PSII-FCPII supercomplex” in our revised manuscript (lines 5-9 in page 18).

As for energy-transfer pathways, most of the structural papers of photosystem-LHC supercomplexes including Pi et al. relied on distances among pigment molecules. Different from such research groups, we have investigated excitation-energy transfer by means of time-resolved fluorescence spectroscopy in the time range of femtoseconds to nanoseconds using the photosystem-FCP supercomplexes. In this study, we discussed excitation-energy-transfer pathways based on time constants of energy transfer as well as pigment distances. Therefore, the present findings provide novel insights into excitation-transfer processes in PSII-FCPII. To emphasize these contents, we added a new paragraph to the Discussion section of “Energy-transfer pathways in the PSII-FCPII supercomplex” in the revised manuscript (pages 18-19).

Reviewers' Comments:

Reviewer #1:

Remarks to the Author:

The authors have addressed all my concerns and the revised manuscript was greatly improved. I have no additional comments and recommend publication of this manuscript.